# Construction of Benzenesulfonamide Derivatives via Copper and Visible Light-induced Azides and S(O)_2_–H Coupling

**DOI:** 10.3390/molecules27175539

**Published:** 2022-08-28

**Authors:** Zhipeng Liang, Ya-Nan Wu, Yang Wang

**Affiliations:** Nantong Key Laboratory of Heterocycles, School of Chemistry and Chemical Engineering of Nantong University, Nantong 226019, China

**Keywords:** aryl azides, arylsulfinic acids, copper/visible light catalysis, S(O)_2_–N coupling, benzenesulfinamide derivatives

## Abstract

We here have developed an S(O)_2_–N coupling between phenylsulfinic acid derivatives and aryl azides by dual copper and visible light catalysis. In this efficient and mild pathway, the reaction produces sulfonamide compounds under redox-neutral condition, which is mechanistically different from the nitrogen nucleophilic substitution reactions. Significantly, this transformation intends to utilize the property of visible light-induced azides to generate triplet nitrene and followed coupling with sulfonyl radicals in situ to achieve structurally diverse benzenesulfinamides in good yields.

## 1. Introduction

Compounds containing sulfonamide core moiety exhibit a wide range of pharmacological activities and a high success rate in therapeutic medicines. For example, cardiosulfa **I** has induced abnormal heart development in zebrafish embryos [1]. MGAT2 inhibitor **II** could have therapeutic potential for the treatment of metabolic disorders [2]. IMB105 **III** has potent in vitro antiproliferative activity against several human cancer cell lines including drug-resistant tumor cells [3]. Sulfa drug **IV** has demonstrated remarkably high anti-bacterial activity against ATCC35218 (Escherichia coli) and ATCC6538 (Staphylococcus aureus) (Figure 1) [4]. Along with these bioactivities, sulfonamide derivatives are also used as meaningful intermediates [5] and functional protecting groups [6] in synthetic organic chemistry. Thus, it may be desirable to develop an efficient approach toward such preparation of benzenesulfinamide derivatives.

The direct construction of S(O)_2_–N bonds could heavily rely on the classical nitrogen nucleophilic substitution reactions. The traditional method is the amination of arylsulfonyl chlorides [7], sodium arylsulfinates [8], or thiophenols [9] with aromatic amines (Figure 1a). Among them, sodium arylsulfinates or thiophenols need to be oxidized before nitrogen nucleophilic substitution process. While generally effective, this synthetic process suffers from the use of aromatic amines as nitrogen sources, which are genotoxic and undesired potential impurities in the synthesis of active pharmaceutical ingredient. Another alternative pathway has been developed with nitroarenes as the starting materials which are inexpensive, readily available, and air-stable compounds that have been widely utilized in synthesis of sulfonamide derivatives in recent years (Figure 1b) [10]. However, it is also limited by the harsh conditions and additional oxidative and reductive agents. It is still difficult to avoid the incompatibility with nucleophilic functional groups. Therefore, reactions with new types of mechanisms for the preparation of sulfonamides are highly desirable. Inspired by previous research on using aryl azides as amino sources [11], we here develop a S(O)_2_–N coupling between S(O)_2_–H compounds and aryl azides by dual copper and photoredox catalysis [12,13,14,15]. This reaction produces sulfonamide compounds under mild redox-neutral conditions, which is efficient and mechanistically different from the nitrogen nucleophilic substitution reactions (Figure 1c).

## 2. Results and Discussion

Organic azides are used as nitrene precursors in metal-catalyzed nitrene transfer reactions [16,17,18], which has the advantages of easy preparation, non-oxidative conditions, and clean reactions (with nitrogen gas as the only side product). Recently, photocatalysts have also shown the ability to activate azides [12]. Based on our previous work, and especially on visible-light photo-catalytic P(O)–N coupling using organic azides as nitrene sources, we initially investigated the reaction of p-tolyl azide **1a** and p-tolyl sulfonic acid **2a** as the model reactants under dual copper and photoredox catalysis conditions. The reaction provided the expected product **3a** in only 12% yield (entry 1). Next, we tested different copper species in this radical coupling transformation. As showed in Table 1, the results indicated that CuCN was optimal in this system with 61% yield target product generated (entries 2–9). Subsequently, we investigated the solvents and photocatalysts for this reaction. With the optimization of different solvents (entries 10–14) and photocatalysts (entries 15–20), the yield could increase to 91% when using CH_3_CN as the solvent and Ir(ppy)_3_ as the photocatalyst (entry 13). Additionally, the reaction could proceed smoothly in the absence of CuCN (entry 21), Ir(ppy)_3_ (entry 22), or Ir(ppy)_3_/CuCN (entry 23). When without visible light, the reaction could not take place under the optimal conditions (entry 24). The reaction was successfully performed at a 1.0 mmol scale, and **3a** was generated in 86%isolated yield (entry 25).

With the optimized reaction conditions available, the substrate scopes of the reaction were explored (see the Appendix A). As showed in Figure 2, a wide range of phenylsulfinic acid derivatives and aryl azides were successfully employed in this transformation. We first examined structurally diverse aryl azides **1** used in this coupling. Initially, we investigated the mono-substituted groups on benzene of aryl azides **1a**–**1j**, which can effectively participate in the reaction regardless of the position of substituents and gave the target products **3a**–**3j** in good yields (61–86%). Without any functional groups on the benzene ring, the reaction produced the **3a** in 61% yield. When introduced electron-donating or electron-withdrawing groups on *para*- (**3b**–**3f**), *meta*- (**3g**, **3h**), or *ortho*-position (**3i**, **3j**) of the aryl ring, the yield increased obviously under the standard conditions. The results indicated that the functional substituents on the benzene ring were significant for high yield benzenesulfinamides generation. With the different substituents on the same position, we found the electron-donating groups could be better than the electron-withdrawing groups. For instance, when the *para*-position with alkyl (**1b**,**1c**) and halide groups (**1d**–**1f**) were used, the methyl and isopropyl substituted benzenesulfinamides **3b**, **3c** generated in higher yields. Additionally, when the *para*- or *ortho*-position with the methyl group on the benzene ring (**1b** versus **1i**), the yield of **3i** decreased to 74%, which could indicate the steric hindrance make a slight influence on this coupling procedure. Unfortunately, when the aryl azides with di-substituted groups on benzene ring, the products **3k** and **3l** were generated in 58% and 35% yields, respectively. Next, we continued to study the application range of arylsulfinic acids **2**. It was gratifying that unsubstituted **3m** and electron-withdrawing substituted **3n** and **3o** arylsulfinic acids could be successfully applied in this transformation and that better yields were obtained, indicating that there was little electronic effect on the reaction.

## 3. Proposed Mechanism

Based on previous experimental work and literature reports [12,19], we proposed a plausible mechanism with three catalytic cycles (Figure 3). Visible-light induces [Ir(ppy)^3^]^3+^ to produce [Ir(ppy)^3^]^3+*^ (an excited photocatalyst Ir(III)*) through energy transfer which participates in two catalytic cycles: a single electron transfer (SET) process with **2** to obtain [Ir(ppy)^3^]^2+^ and a sulfonyl radical **A** accompanied by proton dissociation, and an energy transfer (EnT) process involving **1**, resulting in the loss of N_2_ and the formation of the triplet nitrene **B**. The intermediate **B** was captured by Cu(I) to generate a Cu(III) nitrene intermediate **C**. After underwent a SET process and protonation, **C** was converted to a Cu (II) complex **D** which was then coupled with **A** to give Cu(III) complex **E**. Reductive elimination of **E** formed **3** and regenerated the Cu(I) catalyst. As a minor reaction pathway, intermediate **B** could convert to nitrogen radical **F** directly via a SET process and protonation. Radical coupling of **A** and **F** led to the formation of product **3** in the absence of CuCN (entry 21, Table 1).

## 4. Conclusions

In summary, we have developed a green and economical method for S(O)_2_–N coupling between readily available S(O)_2_–H compounds and aryl azides via dual copper and visible light catalysis under redox-neutral conditions. The reaction not only intelligently utilized triplet nitrene intermediate induced by visible light, but also efficient construction of various structurally diverse benzenesulfinamide derivatives under mild conditions. The further application of nitrene intermediates in biological heterocycles synthesis is still under research in lab.

## Data Availability

Not applicable.

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
