# Peer review of "Construction of Benzenesulfonamide Derivatives via Copper and Visible Light-induced Azides and S(O)2–H Coupling"

_molecules, 2022, doi:10.3390/molecules27175539_

Round 1
Reviewer 1 Report
In this manuscript the authors reported “Benzenesulfinamide Derivatives” by Copper and Visible Light-Induced Aryl Azides and S(O)2–H coupling. Since these scaffolds are biologically and medicinally worthy. The described studies could be an interesting extension of the current knowledge and well established by others. overall, the amount of novelty is not remarkably high. However, I have only few questions that need to be justify before its acceptance.
Comments
1. In scheme 1, authors have to mention previous work and citations as well. Title needs to be changed and language needs to be improved.
2. In line 69 is it 80% yield?? PC should be photocatalyst,
3. Does authors check the reactivity with methoxy and nitro substituted compounds.
4. I can see that authors mentioned in SI file in general info. regarding MPs and IR, but there is no information regarding those.
5. Authors needs to check NMR data with the spectral copies and those are not consistent. For example, compounds 3g and 3l integrations as well as count of protons.
Author Response
Dear Reviewer,
Thank you very much for your valuable suggestions on our manuscript (Manuscript ID: molecules-1864864). We are grateful for the valuable suggestions on how to further improve the manuscript and supporting information (SI). As requested, we herein electronically submit the revised version of the manuscript and SI. The response of comments is attached at the end of the letter.
With these revisions, we hope that this paper is now suitable for publication in MOLECULES. We would like to thank you for your time and effort in handling this manuscript.
With best wishes
Yang Wang
- In scheme 1, authors have to mention previous work and citations as well. Title needs to be changed and language needs to be improved.
Response: Dear reviewer, thank you for your valuable suggestions on our manuscript, we have changed the title in Scheme 1 and improved the language in introduction section.
- In line 69 is it 80% yield?? PC should be photocatalyst.
Response: We have changed ‘80%’ into ‘86%’, and ‘PC’ replaced by ‘Photocatalyst’ in our revised manuscript.
- Does authors check the reactivity with methoxy and nitro substituted compounds.
Response: Dear reviewer, initially, we investigated the electron-donating and electron-withdrawing groups of aryl sulfonic acids in this transformation, respectively. We found both electron-donating and electron-withdrawing groups were suitable for this reaction. However, when we employed methoxy group instead of methyl group, the yield of product was extremely low. When we utilized the aryl sulfonic acid bearing nitro substituent at para-position, the system was very complicated in this reaction.
- I can see that authors mentioned in SI file in general info. regarding MPs and IR, but there is no information regarding those.
Response: We have changed general information in our revised SI
- Authors needs to check NMR data with the spectral copies and those are not consistent. For example, compounds 3g and 3l integrations as well as count of protons.
Response: Thank you for your valuable suggestion, we have checked and revised NMR data in our SI.
Reviewer 2 Report
The manuscript entitled "Construction of Benzenesulfinamide Derivatives Via Copper and Visible Light-Induced Aryl Azides and S(O)2–H coupling" describes the method of synthesis benzenesulfonamide derivatives via the catalytic reaction of aryl azides with arylsulfinic acids. Unlikely that the method proposed by the authors is definitely competitive compared to known methods, since an iridium catalyst is required. However, such a synthetic method has a scientific novelty and may find application. In my opinion, the presented manuscript is acceptable for publication in the Molecules.
In the title of manuscript “Benzenesulfinamide” should be replaced by “Benzenesulfonamide”.
"arylsulfinic acids" should be added to the keywords.
Table 1, line 6: "Tarce" should be replaced by "traces".
Table 1, line 19: "p-Terophenyl" should be replaced by "p-Terphenyl".
In the Supporting Information, it is not necessary to indicate the chemical shift of the doublet in the form of an interval of δ values, for example 7.22 – 7.20 (d, J = 8.0 Hz, 2H) for 3b.
Author Response
Dear Reviewer,
Thank you very much for your valuable suggestions on our manuscript (Manuscript ID: molecules-1864864). We are grateful for the valuable suggestions on how to further improve the manuscript and supporting information (SI). As requested, we herein electronically submit the revised version of the manuscript and SI. The response of comments is attached at the end of the letter.
With these revisions, we hope that this paper is now suitable for publication in MOLECULES. We would like to thank you for your time and effort in handling this manuscript.
With best wishes
Yang Wang
- In the title of manuscript “Benzenesulfinamide” should be replaced by “Benzenesulfonamide”.
Response: Dear reviewer, thank you for your valuable suggestions, we have changed “Benzenesulfinamide” into “Benzenesulfonamide” in our revised manuscript.
- "arylsulfinic acids" should be added to the keywords.
Response: We have added "arylsulfinic acids" in keywords.
- Table 1, line 6: "Tarce" should be replaced by "traces".
Response: We have changed "Tarce" into "traces" in Table 1.
- Table 1, line 19: "p-Terophenyl" should be replaced by "p-Terphenyl".
Response: We have changed "p-Terophenyl" into "p-Terphenyl" in Table 1.
- In the Supporting Information, it is not necessary to indicate the chemical shift of the doublet in the form of an interval of δ values, for example 7.22 – 7.20 (d, J = 8.0 Hz, 2H) for 3b.
Response: We have checked and revised our supporting information.
Round 2
Reviewer 1 Report
The suggested changes has been incorporated in the revised version and I am happy to recommend it for publication.